# Self-Assembled Metal Nanoclusters: Driving Forces and Structural Correlation with Optical Properties

**DOI:** 10.3390/nano12030544

**Published:** 2022-02-05

**Authors:** Sarita Kolay, Dipankar Bain, Subarna Maity, Aarti Devi, Amitava Patra, Rodolphe Antoine

**Affiliations:** 1School of Materials Sciences, Indian Association for the Cultivation of Science, Kolkata 700032, India; mssk2256@iacs.res.in (S.K.); stsm2@iacs.res.in (S.M.); 2Energy and Environment Unit, Institute of Nano Science and Technology, Knowledge City, Sector 81, Mohali 140306, India; dipankar.ra202102@inst.ac.in (D.B.); aarti.ph20257@inst.ac.in (A.D.); 3CNRS, Institut Lumière Matière UMR 5306, Univ Lyon, Université Claude Bernard Lyon 1, F-69100 Villeurbanne, France

**Keywords:** self-assembly, metal nanoclusters, nanoscale forces, structural correlation, optical property

## Abstract

Studies on self-assembly of metal nanoclusters (MNCs) are an emerging field of research owing to their significant optical properties and potential applications in many areas. Fabricating the desired self-assembly structure for specific implementation has always been challenging in nanotechnology. The building blocks organize themselves into a hierarchical structure with a high order of directional control in the self-assembly process. An overview of the recent achievements in the self-assembly chemistry of MNCs is summarized in this review article. Here, we investigate the underlying mechanism for the self-assembly structures, and analysis reveals that van der Waals forces, electrostatic interaction, metallophilic interaction, and amphiphilicity are the crucial parameters. In addition, we discuss the principles of template-mediated interaction and the effect of external stimuli on assembly formation in detail. We also focus on the structural correlation of the assemblies with their photophysical properties. A deep perception of the self-assembly mechanism and the degree of interactions on the excited state dynamics is provided for the future synthesis of customizable MNCs with promising applications.

## 1. Introduction

The study of self-assembly of nanomaterials has been an efficient and powerful strategy in nanotechnology for decades and is still relevant today. Self-assembly is a flexible process where the building blocks spontaneously assemble themselves into a highly ordered large structure through direct interaction and with the help of an external influence [1,2,3,4]. A comprehensive understanding of the thermodynamic forces that drive the self-assembly process is a prerequisite for designing and controlling the morphology of the assembled structure for ideal applications [5,6,7]. The assembly formation offers an enhanced collective property for a wide range of applications [8,9,10,11,12]. The self-assembled architectures with different morphologies have unique directing forces with exclusive photophysical properties. 

However, the structural correlation with the optical properties is limited in the nanomaterials, especially for the nanoparticles (NPs) due to their polydisperse nature and the inherent difficulty in crystallization. Herein, a subclass of NPs, MNCs grabbed attention in self-assembly research due to their atomically precise composition and well-defined structure [13,14,15,16,17,18]. MNCs are ultra-small particles consisting of a controlled aggregation of metal atoms protected by a shell of organic ligands. Such MNCs are usually obtained at the atomic precision (i.e., with a defined number of metal atoms (nM) and ligand molecules (mL) leading to formula (M_n_L_m_)) and display molecular-like properties [15,19,20,21]. The geometry of the clusters must be determined by quantum chemistry methods that often use group theory, and the optic response is described in terms of molecular transitions whose positions and intensities are predicted by sophisticated calculations of quantum mechanics [22,23,24]. Among these molecular-like properties, luminescence has attracted tremendous research interest [25,26,27,28,29]. However, compared with conventional luminescent materials (e.g., organic dyes and quantum dots), the quantum yield (QY) of MNCs is relatively low [30]. Constructing hierarchical self-assembly or inducing aggregation of ligated MNCs has opened the way to boost the luminescence of MNCs [31]. However, uncontrolled aggregation during self-assembly may usually occur due to MNCs’ large specific surface area. On the other hand, the richness of surface ligands in MNCs and the versatile surface functionalities allow for elaborating strategies involving manipulating the driving forces guiding the intercluster interactions. These attractive forces act as a glue between surface ligands of different NCs and can be triggered by external stimuli such as metal ions, pH, macromolecules, solvents, light, etc. MNCs are increasingly used as versatile building blocks to fabricate nanoarchitectures via a self-assembly process. The assembled geometry can be tuned with specific size and composition by controlling the driving forces, templates, or external stimuli [32,33,34,35]. Modulating the nanoscale forces such as dipolar interactions, van der Waals forces, electrostatic interaction, and metallophilic interaction, assembled architectures can be tailored. The correlation between the ligand structure and intra-cluster ligand interaction can play a significant role in fabricating ordered assemblies [36]. The hierarchical assembly of MNCs with atomic precision can be significantly modulated by tailoring the surface ligands [37]. The diverse surface chemistry of the ligands with different moieties provides a versatile weapon for controlling the assembly geometry as well as their photophysical properties [38,39]. A deep understanding of the control of morphology is required to customize the architectures for ideal applications. Hierarchically assembled structures consisting of MNCs are important in nanotechnology because of their collective properties and potential applications in catalysis, electronics, sensors, and storage devices [40,41,42,43,44]. Although most MNCs show weak luminescence, their assembly can enhance the luminescence property based on aggregation-induced emission strategy [30,45,46,47,48,49]. The metal–metal, metal–ligand and ligand–ligand interactions control the relaxation dynamics. The rotational and vibrational motions of the ligands are restricted during assembly formation, and thus the non-radiative channels for relaxation decrease, which enhances the luminescence.

There are many recent reviews on the synthetic strategies to produce self-assembly of NCs and their applications [50,51,52,53,54,55,56,57,58,59]. However, the structural correlation of the MNCs with their optical properties remains unknown, which is also important to design the self-assembled structures with customizable properties. In this review, we focus particularly on the self-assembled structure–optical properties relationship and the driving forces behind assembly formation. The current review article is thus intended to give insights into the fundamental driving forces leading to the self-assembly of metal NCs (copper, gold, and silver) and their impact on photophysical properties. Specifically, we discuss the fundamentals of leading interactions in self-assembled MNCs. The template effect controls the DNA-based self-assembly of MNCs. The use of external stimuli, e.g., light and temperature, controls the assembly process. Versatility in enhanced photoluminescence of self-assembly of MNCs, e.g., QY enhancement and tuning emission color, will be highlighted. Finally, we outline some perspectives on the development of this area.

## 2. On the Structure–Optical Properties Relationship of Metal Nanocluster Assembly

The origin of the photoluminescence (PL), in particular in the near-infrared, from thiolate-protected gold nanoclusters remains elusive. Indeed, it is still a major challenge for researchers to map out a definitive relationship between the atomic structure and the PL property and understand how the metal core (through excitations via the Au(0) kernel) and Au(I)–S surface (through charge transfer excitations) contribute to the PL of Au NCs. The lowest excited states in absorption spectra usually belong to the “core” in nature. In the case of the highest excited states, both “interface-like” and “core-like” (or a combination of both) are involved [60]. The characteristic of such excited states is more “interface-like” (or also called “oligomer band” or “Au−S band”), for which contribution from the Au-S interface in orbitals is more pronounced. Further, electron-rich donor groups from surface ligands may contribute to “surface-like” excited states. In addition, multiple energy transfers associated with intersystem crossings (ISC reinforced) allow an overall boost in PL emission and longer PL lifetimes [61]. The following experimental and theoretical findings were assembled from the literature to derive the energy diagram in Figure 1.

For thiolated metal NCs, many parameters affect how the energy flows following photoexcitation. The main parameters are molecular floppiness, solvent accessibility, and metallophilic interactions, as exemplified with the smallest Au_10_SG_10_ NCs (with SG as glutathione ligand, see Figure 1). Generally speaking, loose and floppy ligand molecules such as glutathione will exhibit several rotational and vibrational degrees of freedom and seriously lower the fluorescence intensity. Solvents, in particular solvent accessibility to the metal core or metal-sulfur interface, may play a role in the deexcitation pathway of NCs. Suppose MNCs are in a high vibrational level of the triplet state. In that case, energy can be lost through collisions with solvent molecules (vibrational relaxation), leaving it at the lowest vibrational level of the triplet state. It can then again undergo intersystem crossing to a high vibrational level of the electronic singlet ground state, where it returns to the lowest vibrational level through vibrational relaxation. Solvent accessibility has been found to play a critical role in the luminescence properties of ligated gold NCs. It is strongly dependent on the nature of ligands and the size of MNCs [62,63]. Aurophilic interactions between interlocked Au(I)-thiolate ring structures for Au_10_SG_10_ catenane NCs [64,65] cause the enhancement of the luminescence for Au(I) oligomers. Usually, the surface ligands are the components that direct aggregation/assembly. Thus, when the assembly of MNCs is produced, there will be a substantial restriction of vibrational and rotational motions of the outer surface ligands of individual MNCs [66]. Building a massive assembly of NCs onto higher-order hybrid superstructures will lead to strong confinement on many length scales. This confinement will also reduce the solvent accessibility, increasing the aggregation-induced emission (AIE) process of luminescent MNC-based assemblies. Finally, intercluster metallophilic interactions also control the aggregations of MNCs and the AIE process. The impacts of the different forces on the AIE process of MNC-based assemblies will be discussed in the following section. 

## 3. Nanoscale Forces on Assembly

Surface interactions can play a vital role in the self-assembly process of MNCs. The structural control and morphological evolution of the directed self-assembly can be controlled by the nanoscale forces such as dipolar attraction, van der Waals interactions, electrostatic interactions, π–π stacking, metallophilic interactions, etc. This section will highlight a couple of examples in which intercluster interactions are the driving force to form assemblies. 

### 3.1. Dipolar Interaction

Numerous building blocks can self-assemble into ordered structures driven by the entropy factor. Ordering the noninteracting hard-sphere results in an entropy increment by having more available free space in the ordered structure compared to the disordered state. Theoretical calculations and simulations predict the most stable ordering of the assembled systems. Self-assembly of particles strongly depends on the particle size and the behavior of the particles based on coulombic, charge–dipole, and dipole–dipole interactions. The dipole moment can originate for the nanocrystals due to the noncentrosymmetric distribution of polar facets or asymmetric lattice truncations [67]. In the case of MNCs, the dipolar attraction between the individual NCs plays a substantial role in structural change. For example, dipolar attraction and van der Waals attraction control the 2D self-assembly of Au_15_DT_15_ into mono, few, and multi-layered sheets [68]. Au_15_ NCs involve dipole–dipole attraction and van der Waals attractions when the solvent is nonpolar. Dipole–dipole attraction arranges the 1D orientation of Au_15_ NCs. The permanent dipole moment (µ) of Au_15_DT_15_ was calculated to be 13.27 D, and the energy of dipolar attraction between two NCs was 3.8 kJ/mol, calculated using the formula [68]
(1)$$E=-\frac{\mu^{2}}{2\pi \varepsilon_{0}r\left(r^{2}-d_{NC}^{2}\right)}$$
where $\varepsilon_{0}$ is 8.85 × 10^–^^12^ C^2^J^−1^m^−1^, *r* is the center to center interdipolar separation, and *d*_NC_ is the diameter of NCs. The 3.8 kJ/mol dipolar energy is higher than the regular dipole–dipole attraction (1.5 kJ/mol). The self-assembly of Au_15_ NCs induced by anisotropic dipole–dipole attraction directs the linear alignment along the x-direction (Figure 2). Further, the inter-NC distance in the self-assembly can be modulated with different solvents resulting in a change in inter-NC dipolar attraction [38]. The dipole moment of Cu_6_ is 0.87 D, corresponding to an energy of 0.02 kJ/mol. Cu_3_ shows a dipole moment of 3.11 D and more energy (0.64 kJ/mol). The combined effect of theoretical and experimental results of dipolar interactions plays a vital role in the self-assembly process and can control the morphology. The advantage of these interactions over van der Waals is that they only provide internal interaction and thus stabilize the assembled structure.

### 3.2. Van der Waals Interactions

Directed self-assembly can be achieved by controlling the van der Waals (vdW) attractions among the NCs. In the MNC assembly, van der Waals forces between the capping ligands of NCs play a pivotal role in forming an ordered structure. The vdW forces are the most ubiquitous nanoscale interaction, originating due to the electromagnetic fluctuations that arise from the continuous movements of opposite charges within the atoms, molecules, and bulk materials. The vdW interactions between the metal cores can be calculated using the formula [69]
(2)$$u_{\mathrm{vdW}}\left(r\right)=-C_{\mathrm{vdW}\;}/r^{6}$$
where *r* is the distance between the atomic or molecular center, and C_v__dw_ is a constant characterizing the interacting species and the surrounding medium. The forces are proportional to 1r6. It is clear that in MNC assemblies, the strength of the van der Waals force is comparable to the strength of other long-range interactions, such as electrostatic interactions. Another approach to estimating the vdW interactions is the additive Hamaker integral approximation [70]. Here, the interacting particles are the sphere, and the dispersion interaction is only present within the particles.
(3)$$U_{vdW}=\frac{A}{3}\;\left[\frac{a_{1}a_{2}}{r^{2}-{\left(a_{1}+a_{2}\right)}^{2}}+\frac{a_{1}a_{2}}{r^{2}-{\left(a_{1\;}-a_{2}\right)}^{2}}+\frac{1}{2}\ln\left(\frac{r^{2}-{\left(a_{1}+a_{2}\right)}^{2}}{r^{2}-{\left(a_{1}-a_{2}\right)}^{2}}\right)\right]$$
where *A* is the Hamaker coefficient and can be calculated using $A=C_{\mathrm{vdW}}\pi^{2}/\;v_{1}v_{2}$ ($v_{i}$ is the molar volume of material *i*) [69].

The morphology of self-assembled Cu NCs capped by 1-dodecanthiol (DT) can be tuned from wire to ribbon (Figure 3A) [71]. The van der Waals attraction between the capping ligands results in assembled nanowires having a diameter of 26 nm (wire-26). The composition of Cu NCs in wire-26 was revealed to be Cu_12_DT_8_Ac_4_. The vdW attraction between two Cu_12_DT_8_Ac_4_ NCs can be calculated using Equation (3) and was found to be 4.1 K_B_T (K_B_ is the Boltzmann constant and T is the absolute temperature), which directs the self-assembly formation of the hydrophobic structures. Here, the use of temperature accelerates the self-assembly formation, whereas the low-temperature kinetically controlled process gives rise to wires like morphology. Prolonged annealing of these wires like geometry leads to reorganization into flower like structures. The thickness can be modulated by annealing pre-assembled wire-26 at 120 °C for 30 min. The thickness is reduced to 1.3 nm, indicating the formation of ribbon-1.3, and the composition changes to Cu_8_DT_8_. The addition of DT facilitates the conversion of the wire into a ribbon by increasing the extent of compactness and arrangement order of the self-assembly. Here, the driving force is mainly vdW attraction, which enables the reorganization of the self-assembled NCs, and the energy is calculated to be 3.8 K_B_T between the Cu_8_DT_8_ NCs. The flexibility of the alkyl chains is more dynamic at high temperature and thus redirected into a compact and ordered structure. As both systems are composed of a self-assembly strategy, the ribbon and wire are highly luminescent. The driving force behind the 1D orientation of self-assembled Cu NCs into nanowires is firmly dipolar attraction [71].

We will discuss another example where Au NCs protected with hydrophobic alkyl thiol are self-assembled into single-cluster-thick sheets through vdW and intercluster hydrophobic–hydrophobic attraction [72]. The self-assembly and reorganization are due to the slight difference between two miscible solvents, BE and LP (dibenzyl ether and liquid paraffin, respectively) (Figure 3B). Solvent evaporation is enhanced after annealing at 140 °C, and Au_15_ clusters aggregate [72]. The elevated temperature to enhance the mobility of the DT alkyl chain and the hydrophobic attraction between two NCs is also found to be 7 K_B_T. This strong interaction leads to the formation of the aggregate into nanosheets. The interfacial energy at the LP-BE surface also drives the reorganization process to form single-cluster-thick sheets. Therefore, by choosing stabilizing ligands as the capping ligand for MNCs or appropriate solvents, van der Waals interactions can be controlled to guide the self-assembly processes. 

### 3.3. Electrostatic Interaction

The electrostatic interactions between the NCs can primarily control the formation of directed self-assembly. Electrostatic interaction is almost as ubiquitous as van der Waals interactions. The unique property is that electrostatic interaction can be attractive, repulsive or directional, whereas van der Waals interactions are attractive in nature. The charge of the overall NCs or alteration of the surface capping groups introduces variation in self-assembly morphology [73]. Several electrostatic interactions such as C–H···π interactions, agostic interactions, H– bonding, and π–π stacking between the NCs can direct the formation of these arrays. The structure of the NCs can be also modulated by the change in symmetry of the ligand environment. Though the Au_246_(p-MBT)_80_ NCs [p-MBT = p-methylbenzenethiol] have nearly spherical geometry, they do not prefer cubic arrangement; instead, they formed lower symmetry monoclinic arrangement [74]. This monoclinic packing is driven by the intermolecular C–H···π interaction, maximizing the van der Waals interaction between the ligands. It is essential to match the symmetry to maximize the interaction of surface ligands in ordered assembly. In this case, the p-MBT ligands self-organize amongst themselves in a highly ordered manner. Twenty-five of the surface ligands at the pole site are arranged into four pentagonal circles (Figure 4A). Along with this α-rotation, there are β-parallel patterns at the waist site having six p-MBTs arranged into three alternating parallel pairs. The highly ordered surface p-MBTs are found to interact between the C–H bonds from the phenyl moiety or the –CH_3_ groups and the ᴨ-electrons. This intermolecular C−H···π interaction specifically stabilizes the 25 p-MBT ligands at the pole as well as ligands of the β-parallels (Figure 4B) [74]. Another exciting phenomenon was discovered by Tang’s group where Au NCs self-assembled into nanocubes with uniform body-centered cubic (BCC) packing and generated unique properties different from the individual NCs [75]. The chirality of Au NCs was changed by simply changing the chirality of the capping ligand from R to S-Tol-BINAP. Au_3_[R-Tol-BINAP]_3_Cl and Au_3_[S-Tol-BINAP]_3_Cl NCs [(*R*)- or (*S*)-2,2′-bis(di-*p*-tolylphosphino)-1,1′-binaphthyl] were prepared, and the Au atoms experienced strong aurophilic interactions amongst each other (Figure 4C) [75]. The DLS study reveals that the clusters were in a dispersed state when 40% n-hexane was used, and after that, aggregates started forming. With 60% n-hexane, the aggregates were irregular and amorphous, becoming uniform nanocubes with 70% n-hexane. From the crystal structural analysis of the self-assembled nanocubes, it was found that each Au cluster faced six nearby Au clusters where three Au clusters were involved in C−H···π interaction with the central Au one forming both outward and inward ring pairs. Another three Au clusters also experience C−H···π interactions, and the energy was ~1.5 to 2.5 kcal/mol, which is much higher than the molecular thermal energy (0.57 kcal/mol). 

Self-assembly of Cu_7_ NCs from rods to platelets to ribbon-like structures can be directed with three different positional isomers of the surface capping ligand (Figure 5A) [76]. From the matrix-assisted laser desorption ionization mass spectrometric study, it is clear that the steric hindrance of the ligand is responsible for morphological evolution, keeping the individual building block (Cu_7_ NCs) constant for all the assemblies (Figure 5B). Although the composition is identical, the building blocks experience the different extent of agostic interaction (Cu···H–C) for three different isomers of the dimethylbenzenethiol (DMBT) ligand depending on the position of the –CH_3_ group. The interplay between the extent of agostic interaction and π–π stacking between the NCs directs the shape from rod to ribbon. Figure 5C depicts the schematic representation of ribbon like assemblies from Cu_7_ building blocks where the extent of the agostic interaction is the maximum. With these structural changes, optical properties and excited state relaxation dynamics of the red phosphorescent assemblies are also differently correlated with their degree of compactness (Figure 5D,E).

Xie et al., obtained customizable shapes of Ag_44_(p-MBA)_30_ (p-MBA = para-mercaptobenzoicacid) nanocluster super crystals (SCs) by changing the counterions and their concentration of counterions [77]. The geometry changes from a lower symmetry rhombohedral (D_3d_) to a higher symmetry octahedral (O_h_) by altering H^+^ with non-H^+^ counterion (Cs^+^). Again, enhancement of the Cs^+^ concentration leads to tailoring the geometry from O_h_ to concave O_h_. The growth kinetics of the SCs manipulate the shape and electrostatic interactions, which follow the charge-screening-assisted nucleation mechanism. Thus, we can conclude that the degree of electrostatic interaction is a primary decisive parameter leading to aggregate formation with the oppositely charged systems. One of the recent studies by Pradeep’s group showed that simple hydrogen bonding could also direct the self-assembly of NCs using the previously discussed ligand p-MBA. Gold nanorods were functionalized with p-MBA, and Ag_44_ NCs were allowed to interact with this GNR@p-MBA, which resulted in directed self-assembly (Figure 6A) [78]. Ag_44_ NCs form assembly through H-bonding with carboxylic acid dimerization of the capped surface ligands. Pradeep et al., investigated the factors directing the morphological evolution. This study suggested that the p-MBA molecules functionalized on GNR can interact with the surface capped ligands of Ag_44_ NCs, leading to a decrease in the extent of H-bonding interactions between the ligands of two adjacent NCs. Self-assembly of Ag_25_ NCs capped with p-aminothiophenol (PATP) into uniform and highly ordered lamellar silver nano leaves (NLs) was studied by Wang’s group [79]. The main driving forces to direct the self-assembly process are electrostatic or covalent bonding and π–π stacking. A couple of studies confirm that the PATP exists in a quinonoid model where the –NH_2_ (electron-donating group) and the –SH (electron acceptor group) are connected by a conjugated benzene moiety. The Ag NCs are bound with these –SH and –NH_2_ groups of the PATP ligand existing in quinonoid form (Figure 6B). The smallest building blocks for this lamellar assembly are [Ag_25_(PATP)_6_]^6+^, and they are interconnected as Ag_25_-PATP-Ag_25_. Now, π–π stacking interaction between the adjacent benzene rings of the building blocks leads to the formation of uniformly ordered silver NLs. To minimize the steric repulsion and entropy, head-to-head assembly is formed with the help of dipole–dipole interactions, whereas the π–π stacking having the temperature and solvent effect leads the side-by-side organization process.

### 3.4. Metallophilic Interaction

The M···M bond in self-assembly is attributed to the weak attractive interaction between the electrons in the filled d^10^ or pseudo-filled (d^8^) orbital of the metal. The manifestation of metallophilic interaction is crucial for the self-assembly of many NC compounds. Metallophilic interactions have been observed for the late transition metals, more prominently group 11 M(I) salts [80]. Metallophilicity directed assembly has recently turned into an emerging area of research, and several studies are reported based on aurophilic (Au^I^···Au^I^), argentophilic (Ag^I^···Ag^I^), and cuprophilic (Cu^I^···Cu^I^) interactions [41,81,82]. Metallophilic interaction in the shell is a decisive parameter for the luminescence of NCs [83]. Xie et al. designed a colloidal self-assembly of [Au_25_(SR)_18_]^−^ NCs into nanoribbons using aurophilic interaction (Au^I^···Au^I^) (Figure 7) [82]. The self-assembly is initiated by the surface motif reconstruction of NCs from the pristine SR-[Au^I^-SR]_2_ motifs to longer SR-[Au^I^-SR]_x_ motifs (x > 2) along with a partial expense of Au^0^ species from the Au_13_ core. This is confirmed by XRD patterns of randomly aggregated [Au_25_(p-MBA)_18_]^−^ NCs and XPS of Au 4f (Figure 6). The XPS spectra show an increase in Au^I^ species accompanied by Au^0^ reduction in the self-assembled nanoribbons (Figure 7). The Au L_3_ edge X-ray absorption near edge structure (XANES) and FT-EXAFS showed a distinct increase in the content of Au-S and Au_staple_−Au_staple_ characters, which supports the formation of long SR-[Au^I^-SR]_x_ motifs. The aurophilic interaction is the driving force for the 1D arrangement of Au NCs into nanoribbons. Aurophilic interaction (Au^I^···Au^I^) plays a crucial role in switching fluorescence to phosphorescence. A small-angle X-ray scattering (SAXS) study confirmed the presence of Au-Au interaction, which drives the self-assembly process and also determines phosphorescence. Bakr et al. reported that argentophilic interactions are crucial for the stability of the silver cluster skeleton [84]. From the single crystal analysis, the Ag–Ag distance is shorter than the van der Waals radius (~3.44 Å), which indicates argentophilic interactions in 0D NCs of the Ag_16_ skeleton. Further, with the employment of a chloride template, 4,4^′^-bipyridine linker, controlled assembly of 0D, 1D, and 2D NCs based framework can be achieved. The importance of argentophilic interaction in directing the self-assembly is also reflected in a study by Xin et al. [81]. Here, Ag_9_ NC constructed a nanofibre structure by interacting with succinic acid. Argentophilic interaction (Ag^I^···Ag^I^)-directed assembled structure was obtained with aggregation-induced emission. Zhang et al. reported the self-assembly of Cu NCs capped by 1-dodecanethiol with enhanced luminescence. This significant enhancement in PL largely depends on the NC interactions (Cu^I^···Cu^I^) and the rigidity of the capping ligands. In this case, two different cuprophilic interactions exist, intra and inter cuprophilic interactions between two NCs. In the case of ribbons, the loose arrangement of NCs results in weaker cuprophilic interaction [41]. The control in structural geometry from sheet to ribbon depends on the extent of cuprophilic interactions. Experimental findings and computational studies declare that the metallophilic interactions are weak and are surpassed by strong electrostatic interaction or solvent molecules. Although several studies of self-assembled MNCs have been reported based on metallophilic interaction, the strength and nature of these interactions in MNC assembly are poorly understood. 

### 3.5. Amphiphilicity of NCs

Amphiphiles are natural or synthetic molecules that can form self-assembled micelles, nanotubes, nanofibers, lamellae-like structures, etc. Amphiphiles are composed of hydrophilic and hydrophobic components. Despite having weak forces, it helps the overall structure become more flexible toward the formation of assembly. Hydrogen bonds are essential for the amphiphiles to gain stability in solution, and hydrophobic interaction is the second driving force to form an assembly [85]. The hydrocarbon amphiphiles can be used widely in nanotechnology, self-assembly of noble metal NCs, and also in the surface modification of the NCs. For example, the self-assembly of amphiphilic Au_25_(MHA)_18_@xCTA NCs (MHA = 6-mercaptohexanoic acid, x = 6–9 and CTA= cetyltrimethylammonium) into stacked bilayers with regular interlayer packing can be observed at the air–liquid interface (Figure 8A) [86]. This was formed by the ion-pairing reaction between CTA^+^, hydrophobic cation, and −COO^−^, derived from the anionic carboxylate group of the hydrophilic NCs. The potential amphiphilicity of the NCs is observed because of the coexistence of hydrophilic MHA and hydrophobic MHA····CTA ligands on the NC surface as well as the flexible chain structure of the surface ligands. Zeng et al. reported the self-assembly of metal NCs guided by the electrostatic repulsive force among surfactant molecules and metal halides. Au NCs form a hierarchical rod/tube-like assembly, whereas giant vesicles and dandelion-like assembly can be found for Pt NCs. Pd NCs and PdS NCs self-assemble into rhombic/hexagonal platelet-like structures (Figure 8B) [87]. Hao et al., in their study, showed an enhancement of fluorescence intensity with the formation of self-assembled amphiphilic Cu NCs [88]. The amphiphilic complex of GSH-capped Cu NCs was prepared by electrostatic interaction and then assembled in an aqueous solution in the presence of a surfactant by rearranging the surface ligands. GSH-Cu NCs are the building blocks that self-assemble in the presence of hydrophilic and hydrophobic moieties, resulting in supramolecular architectures.

## 4. Template-Directed Assembly

In the above section, we have discussed the assembly induced from surface ligands; templates such as DNA, polymers, and macrocycles can also influence the self-assembly process. Templates are a substrate with an active site to bind with the NCs. The variation in templates can result in different assembled morphologies by controlling the interaction in the hybrid assemblies. We will discuss some recent progress on template-directed assembly that entirely depends on template-NC interaction.

### 4.1. DNA Template-Directed Self Assembly

DNA-templated MNCs are a new generation of functional materials, which have widespread applications in catalysis, sensing, bio-imaging, and therapeutics [51]. This section will demonstrate the structures and optical properties of DNA-functionalized MNCs. DNA is one of the emerging materials for constructing supramolecular assemblies [89]. Highly fluorescent and color-tunable Ag NCs can be synthesized by choosing a proper sequence of DNA templates. Martinez and co-workers synthesized bright emitting Ag NCs by using different arrangements and lengths of single-strand DNA (namely, Ag1, Ag2, Ag3, and Ag4); for example, green fluorescent Ag NCs (λ_em_ = 550 nm) were synthesized by using the Ag1 sequence. Similarly, the emission of the Ag NCs was tuned to 600 nm (orange), 650 nm (red), and 700 nm (NIR) using other DNA sequences Ag2, Ag3, and Ag4, respectively (Figure 9A). The stability of these Ag NCs is mainly dictated by the sequence and the length of DNA [90]. NIR-emitting Ag NCs (Ag4) are most stable under salt conditions compared to the other three NCs because Ag4 has a longer nucleotide length, which prevents the salt-induced aggregation of the NC core.

Qing et al. synthesized ultrasmall-sized fluorescent copper NCs using poly-thymine as a template [91]. First, they prepared the DNA-Cu^2+^ complex with sequence-specific DNA in MOPS buffer and then reduced it to Cu NCs. Different types of single-stranded DNA were used as templates, namely, single-stranded DNA (ssDNA), poly adenine (poly-A), ploy thymine (poly T), ploy guanine (poly G), and poly cytosine (poly C). Only the Cu NCs templated by poly T emitted fluorescence at 615 nm under the excitation of 340 nm light. In contrast, no such fluorescence was observed for other sequences (poly A, G, and C). The fluorescence intensity of Cu NCs increased as a function of the length of poly T. Moreover, the size of Cu NCs was dictated by the length of poly T of the ssDNA template. Furthermore, they used different types of T-rich domain ssDNA (40 mer). They found that a particular sequence domain produced intense fluorescent Cu NCs. Thus, it was concluded that T polymerization has an important role in forming fluorescent Cu NCs (Figure 9B). NC-based self-assembled structures exhibit more compactness, strong fluorescence, and excellent photostability than bare NCs. Willner and his co-worker reported self-assembly Ag NC-polymerized nanowires using a sequence-specific DNA template [92]. Two different emission colors Ag NCs were prepared, yellow emission (λem = 570 nm) and red emission (λem = 635 nm) (Figure 9C). Self-assembled NCs are seen from AFM and confocal fluorescence microscopy (Figure 9D). The length of the red-emitting self-assembled Ag NCs is in the micrometer order.

### 4.2. Linker-Directed Assembly

The intrinsic attractive interactions among themselves generally drive regulated self-assembly of molecular NCs, leading to large superstructures. However, small thiolate or non-thiolate ligands that can provide coordination sites to the NCs can control the inter-NC assembly. Ackerson et al. reported a diglyme linker’s dynamic assembly of Au_20_(PET)_15_ NCs [93]. The close spatial proximity between the NCs core via weak diglyme oxygen-Au interaction caused electron delocalization in dimeric Au_20_-diglyme-Au_20_. On the flip side, thiolate ligands can covalently attach to the Au core to form a dimeric and trimeric assembly of NCs, which are well-separable by chromatographic techniques [94]. The multimers of plasmonic Au_~250_ NCs exhibit additional transitions in their absorption spectrum due to hybrid LSPR modes. The assembly of NCs via coordinating molecules can be rationally extended to the metal-organic frameworks (MOFs), also referred to as cluster assembled materials (CAMs), with enhanced functionalities and improved photophysical properties. Suitable organic ligands can bind and spatially separate the NCs to form a three-dimensional rigid framework. Luminescent dodecanuclear Ag_12_ NCs form rigid NC-based MOFs by coordinating linear 4,4′-bipyridine (bpy) ligands [95]. The incoming bpy ligands replace the terminal CH_3_CN molecules from Ag_12_ NCs and connect the Ag–S NC-based nodes, leading to a porous coordinating framework with yearlong stability in contrast to the few minutes of stable-isolated NCs (Figure 10). Moreover, the MOF exhibits enhanced QY (12.1%) and ultrafast dual-functional fluorescence switching with turn off by O_2_ and turn on by volatile organic molecules, which are advantageous in electronic and sensing applications. Solvent-induced transformation of Ag_12_ occurs in the presence of a bpy linker to generate silver CAMs with blue-red dual emission at low temperature [96]. In addition, the mixed linker approach leads to a framework with improved luminescence and thermochromic properties. The chiral [Au_1_Ag_22_(S-Adm)_12_]^3+^ superatom serves as the building blocks of a 3D framework with anionic forces [97]. The superatoms are packed into a racemic mixture and enantiomerically pure crystals via intermolecular Ag-F bond formation depending on the linker concentration. The pure enantiomers display strong circularly polarized PL, whereas the racemic crystals show none. In another approach, pre-synthesized GSH-capped Au NCs were incorporated in ZIF-8 by coordination-assisted self-assembly. The spatial distribution of NCs in the framework restricted the ligand rotation and resulted in a 10-fold increase in electroluminescence compared with aggregated Au NCs. The linker-directed assembly of superatomic NCs has laid a foundation for further development in CAMs with diverse functionalities and desirable physical properties.

## 5. Guided Assembly by External Factors

The use of external influences and fields to control the assembly process has long been a powerful method for tailoring the morphology and optical properties of the metal NCs. The self-assembly of metal NCs influenced by external factors such as light and temperature will be discussed here. The self-assembly of metal NCs can be observed by the photoactivation of NCs, with the temperature change, or by coordinating the NCs with organic species. 

### Light-Triggered Self Assembly

The light-induced self-assembly of metal NCs is an emerging field of research that can be achieved using photoactive ligands such as azobenzene and spiropyran. The critical factor of photoisomerization is extensively studied for these ligands due to the tendency to change their conformation upon irradiation of a particular wavelength of light. In a recent study, self-assembled Cu_3_ NCs were prepared using an azo-group-containing ligand, which is highly light-sensitive. The straight self-assembled nanofibers bend upon exposure to UV light instantly (Figure 11A) [98]. This structural transformation is due to the conformational change of the azo-group-containing ligand from trans to cis in light. This isomerization of the peripheral azo-groups in the Cu_3_ NCs results in disassembly. Besides the isomerization process, solubility also plays a vital role. The straight nanofibers were fabricated in a binary solvent mixture of toluene and methanol, but the cis conformation was more soluble here. Thus the solvophobic interaction in trans-isomer favors the aggregation process. The light-triggered self-assembly can also be explained by taking an example of phenylethanethiol-tethered Au_25_ NC, which is stapled by the photoactive spiropyran (SP) (Figure 11B) [99]. T. Udayabhaskararao et al. observed that synthesized NCs show photoisomerization in a reversible manner when exposed to UV and visible light due to the dipolar–dipolar interaction (Figure 11C). They demonstrated that the SP-functionalized NCs initially formed in closed rings exhibited no UV absorption peak in the 550–600 nm region. However, exposure to UV light resulted in isomerized into an open ring form, i.e., merocyanine with an absorption band at 587 nm, and the color changed from yellow to purple. The self-assembly of NCs will separate again in a closed ring form if incubated in dark or visible light. Therefore, the assembly–disassembly can be obtained several times for the SP-functionalized NCs; using this strategy, light-induced assembly was found in NCs. Similarly, using the Brust Schiffrin single-phase method, Rival et al. synthesized monothiolated azobenzene-protected Au_25_ NCs showing light-triggered reversible self-assembly—disassembly when irradiated with UV light (365 nm) and visible light (435 nm) [100]. This study shows an attractive disc-like superstructure was obtained due to self-assembly of [Au_25_(C_3_-AMT)_18_]^−^ NCs. The photoactivation of NC in the presence of light results in long-range self-assembly and is analyzed using atomic force microscopy AFM, DLS, TEM, and subsequent ET reconstruction. Apart from this, they also synthesized phenylethanethiol (PET)-capped gold NC, with no photoswitching effect. Secondly, the synthesized PET analog [Au_25_ (PET)_18_] NC obtained the same TEM images of individual NCs before and after the light-induced assembly. Here, the attractive dipole–dipole interaction was the driving force for the formation of superstructures of NC upon illumination of light. 

Zhang et al. synthesized thiolated azobenzene-capped Cu NCs with light-triggered self–assembly phenomena in a different study [101]. The formation of long nanoribbons of Cu_12_DT_8_Ac_4_ NCs was due to the observed permanent dipole moment. The combination of dipolar and van der Waals attraction under UV light formed a spherical superstructure. Here, the light was used as an external stimulus for the reversible self-assembly.

Coordination between the charged MNCs and ions with opposite charges can also direct the self-assembly process. Xie et al. reported an assembly formation via the electrostatic interaction between the negatively charged MNCs in water with the divalent cations, e.g., Cd^2+^ and Zn^2+^, added externally in the solution [102]. This assembly showed high orderliness and regularity and enhanced PL, implying a strong synergistic effect between the NCs. Another study of zinc-mediated self-assembly was reported by Chattopadhyay et al. [103]. This report investigated programmable assembly formation between the mixed ligand (L-histidine and mercaptopropionic acid)-capped Au_14_ NC and Zn^2+^. Anisotropic assembly was designed with Ag_52_ and Ag_76_ NCs using tBuC_6_H_4_SH and MeOC_6_H_4_SH and a bis-(diphenylphosphino) methane ligand [104]. This study revealed that the ligands’ regiospecific distribution and arrangement formed the anisotropic growth. A very recent study reported the formation of chiral assembly from achiral, atomically precise Ag_9_ NCs [105]. Here, a second metal ion, Ba^2+^, facilitated the assembly process, and thus nanotubes with chiral cubic lattice were obtained. These hierarchically organized self-assembled structures formed chiral hydrogels with enhanced luminescence in an aqueous solution. Other external stimuli on which the self-assembly process largely depends are pH and the solvent. As these parts were focused on in many previous articles, we do not discuss them in detail [4,51,52]. We will discuss a couple of examples regarding these factors. Tan et al., in their study, synthesized a self-assembled structure from thiosalicylic acid-capped silver NCs, which showed nanofiber-like morphology. The assembled geometry can be tuned from nanofiber to amorphous morphology with the solvent change from THF to toluene [106]. A diphosphine-protected [Au_6_]^2+^ cluster exhibited aggregation in an aqueous organic solvent with an appropriate water content. In a specific range of water, the aligned cluster assembly showed a J-aggregate-like absorption response and PL enhancement [107]. Liu’s group used a pH-guided assembly strategy to fabricate protein/Cu NC hybrid nanostructures with stable and bright luminescence properties [108]. The aggregates showed strong AIE behavior at pH 3.0, but the high reversible pH-responsive nature of Cu NCs showed weak luminescence at pH < 1.5 or >4.0, and the aggregates were soluble. These strategies are used to design some new architectures with unique properties for specific applications.

## 6. Optical Properties of Self-Assembled Nanoclusters: Aggregation-Induced Emission 

Luminescent MNCs are highly in demand because of their various application possibilities in bioimaging, biosensing, fabrication of LEDs, etc. However, MNCs with sub 2 nm size have very low QY, limiting their practical applications [109,110]. The luminescence property can be improved by aggregation-induced emission strategy, which Tang’s research group first discovered [111]. Self-assembly is an efficient strategy for achieving tunable emission and customizable shape and size of the ordered structures by modulating the spatial distribution between the building blocks. For example, by controlling the inter NC distance between the individual Cu NCs, Zhang et al. tuned the photophysical behavior significantly [38]. The inter-NC distance is controlled by changing the solvents with different dielectric constants. The radiative relaxation from the Cu-centered triplet state influences the charge transfer from ligand to the Cu core, and it changes the emission color. This study reveals that the solvents can regulate the aggregation process effectively along with their luminescent behavior. However, the luminescence property of the MNCs depends both on the metal core and the surface capping ligands (see Section 2). Though many works have been done by restricting the rotational and vibrational motion of the capping ligand to alter the self-assembly-induced emission, very few studies reveal the effect of the metal defects on AIE. Zhang et al. demonstrated the effects of metal defect states in their self-assembly of dodecanethiol (DT)-capped Cu NCs (Figure 12A) [112]. The metal defect states speed up the self-assembly process by coordinating with Cu and changing the surface properties of self-assembled nanosheets. The enhancement in absolute QY originated from a triplet state (T_2_) related to metal defects. Figure 12B depicts that T_1_ is the original triplet state determined from ligand to metal–metal charge transfer (LMMCT), whereas metal defects largely influence T_2_, thus providing a lower energy level. This phenomenon allows the relaxation to occur from the T_2_ state to the intermediate state attributed to the metal defects. As a result, they observed a red-shifted emission from 490 to 550 nm with an enhanced QY of 15.4%. Here, the increased Cu(I) facilitates the radiative relaxation process. Based on this strategy, Au(I) was doped onto the Cu NC-self-assembled nanosheets to influence the LMMCT process deliberately [113]. Doping with the Au atom enhances the QY and results in red shifting of the emission. This result is attributed to the Au(I)–Cu (I) metallophilic interaction-directed charge transfer from Cu to Au. Doped Au on Cu self-assembled nanosheets generates a stable Au(I) centered state, which leads the charge transfer from the ligand to Cu and then from Cu to Au. Therefore, a red-shifted emission spectrum is observed at ~600 nm with a longer lifetime, as the Au(I) doping lowers the energy. 

In addition to metal defect states, metallophilic interactions are essential to produce self-assembled NCs with significant luminescence enhancement. A controlled and ordered cuprophilic Cu(I)···Cu(I) interaction assisted Zhang’s group to report assembly-induced emission [41]. They designed a blue–green emissive assembly, and the emission peak was centered at 490 nm with a QY of 6.5%. The assembly-induced emission greatly depended on inter and intra-NCs cuprophilic interactions and the rigidity of the capping ligands (Figure 12C). Firstly, the cuprophilic interactions influenced the radiative relaxation, and secondly, the non-radiative relaxation decreased upon the restriction of the vibration and rotation of the ligands. Although the highest QY was achieved with self-assembled ribbons, they also prepared loosely bound self-assembled sheets with a reduced QY of 3.6%. This result signifies a strong correlation between the cuprophilic interaction and emission enhancement. Based on this strategy, another self-assembled Ag_9_ NC was also prepared, promoting the implementation of AIE [81]. Xin’s group prepared Ag_9_(MBA)_9_ NCs, which produce luminescent hydrogels in interaction with succinic acid. Several non-covalent interactions along with argentophilic interaction are responsible for the assembly-induced phosphorescence emission. The phosphorescence is said to originate from a triplet state attributed to argentophilic interaction with LMMCT. On hydrogel formation, the ligand’s rotation becomes suppressed, which lowers the energy of the triplet state and yields bright orange–red emission, whereas for solution-state Ag_9_ NCs, the surface ligands are free to rotate or vibrate, and thus non-radiative relaxation channels increase. The self-assembled highly phosphorescent nanofibres can be used as a phosphor LED.

Another fundamental question may arise between the M(0) core and M(I)-SR: which is more efficient for the PL enhancement? Several studies have revealed that metalcore solely does not determine the PL properties, as NCs with the same no. of metalcores but different surface capping ligands result in various PL phenomena. Xin et al. discovered an exciting finding based on the AIE strategy where the non-luminescent oligomeric Au-(I)-thiolate complexes became highly luminescent upon aggregation. The luminescence originated from a metal-centered triplet state, and it depended strongly on the degree of aggregation. The increased aurophilic interaction in the denser aggregation enhanced the luminescence by reducing the non-radiative relaxation probability of the excited states [114]. Furthermore, Xie et al. synthesized a weakly emitting gold NC with a QY of 7%; 60% luminescence enhancement occurred by rigidifying the Au(I)-thiolate shell of NC with cations due to aurophilic interaction [115]. Most importantly, the rigidity of the gold shell by binding with bulky groups resulted in a drastic reduction of the non-radiative channels.

The change in core composition also influences the PL properties. For example, Xie et al. developed Au@Ag NCs where Ag(I) acts as a bridge between slight Au(I)-thiolate motifs and generates aggregation-induced emission [116]. The relative QY was found to be 6.8%, with a strong red emission at 667 nm. This luminescence was generated from the metal-centered triplet state affected by the Au(I)/Ag)(I)-thiolate surface state. In another study, an anti-galvanic exchange reaction achieved bimetallic NCs with strong emissions. Bain et al. synthesized MPG capped Au NCs, which exhibited very weak red fluorescence at 735 nm, whereas the bimetallic AuAg NCs showed bright luminescence at 674 nm (r-AuAg) and 574 nm (y-AuAg) with different Ag concentrations (Figure 13A). In this study, aggregation-induced emission strongly depended on the length of surface motifs, and this impacted the photophysical properties of AuAg NCs. Further, the solvent played a vital role in engineering the surface motifs of the NCs, and accordingly, 13-fold PL-enhancement was found for y-AuAg NCs. The TEM images confirmed that circular assembled structures were formed with a 60% water-EtOH mixture (Figure 13B). Therefore, the PL intensity increases with the degree of aggregation. Further, the excited state lifetime study revealed that the Au-thiolate motifs are more stable in a less polar solvent, which causes the enhancement of emission intensity (Figure 13A) [117]. Another solvent polarity-directed controlled assembly of NCs was employed by our group previously to improve the PL QY as well as tune the emission color [118]. The red luminescent Cu_34-32_(SG)_16-13_ NCs showed very weak emission at ambient temperature but bright-red emission upon freezing (temperature < 0 °C). The restriction of molecular motion in the frozen state may be attributed to the emission enhancement. With increasing EtOH from 0 to 90% (Figure 13C,D), the particles with low surface charge suffered from a loss of stability in the solution. Therefore, the particles came closer due to inter and intra Cu(I)-Cu(I) cuprophilic interaction. The Cu NCs in an aqueous medium undergo fast internal conversion from S_n_ to S_1_ and subsequently to a triplet state (T_1_) (Figure 13E). Further, this comes down to a low-lying triplet state T_2_ and undergoes radiative relaxation to the ground state (Figure 13E). The enhancement of intensity after aggregation is due to the restriction of the rotational and vibrational motion of the ligands. Here, the radiative decay occurred directly from high-lying T_1_, which resulted in blue shifting of the emission maxima from 625 to 597 nm.

The self-assembly of Au_3_[R-Tol-BINAP]_3_Cl drastically changes the optical properties compared to the non-luminescent Au NCs. The self-assembly exhibits an orange emission centered at 583 nm, and a significant stoke shift (~138 nm) suggests phosphorescence, originating from a triplet state LMCT or LMMCT. These assemblies also show strong circularly polarised luminescence (CPL) response. The PL intensity is found to display enhancement with the increase in the n-hexane fraction, and the maximum intensity is found with 70% n-hexane (QY ~3.6%). Structural and morphological characterization was brought into play to decode the factors behind such behavior. Concentration-dependent ^1^H NMR spectroscopy revealed that the strong C–H···π interaction amongst the assembled Au NCs limits the intramolecular rotation of the chiral ligands. Consequently, the non-radiative channels for relaxation are closed, and the radiative decay dynamics increase drastically [75]. The aggregation strategy induces the enhancement of the luminescence property and can change the origin of emission. Konishi et al., in their study, displayed the switching of emission type from fluorescence to phosphorescence upon aggregation of [core + exo]-type [Au_8_]^4+^ clusters [119]. Monomeric NCs show fluorescence, whereas their aggregation displays phosphorescence-type emission, primarily due to the closed packing of NC assemblies. The phosphorescence can be observed only at a particular alignment of the NC orientation. 

The electronic interaction between NCs in the assembled structures dramatically influences the dynamics at an ultrafast time scale. Dimeric Au_20_-diglyme-Au_20_ showed different relaxation behavior compared to Au_20_-diglyme [93]. Circularly polarized pump pulses can differentiate between low-spin monomer and high-spin dimeric species, as Au_20_-diglyme-Au_20_ exhibits additional picosecond decay due to the spin-fli p relaxation process. Knappenberger, Jr. et al. later investigated the effect of the length of bridging molecules on the electronic transitions of dimeric assemblies [120]. The increase in the dimer-specific electronic relaxation time with the decrease in the length of bridging molecules reveals the presence of intercluster distance-dependent electronic states. An unprecedented appearance of electron–phonon coupling, which is the characteristic of plasmonic NPs, is observed in the one-dimensional assembly of Ag_4_(S-Adm)_6_ building units [121]. The pump power-dependent exciton relaxation dynamics indicate that the CAMs exhibit molecular-like and plasmonic behavior. Further investigation into the ultrafast dynamics of self-assembled materials will unveil the detailed electronic coupling owing to inter-NC interactions. These discussions mentioned above manifest a solid correlation between the structural and optical properties of the self-assembled NCs. Tailoring the surface motifs to fabricate architectures with customized composition, length, and shape synchronously engineer their optical properties. Indeed, in addition to enhance the emissive properties of NCs through the self-assembly formation, the use of chiral NC leads to strong circular dichroism intensity and a remarkable circularly polarized luminescence response [70].

## 7. Conclusions and Outlook

In summary, this review article has highlighted MNC-assemblies of different sizes, shapes, and compositions, providing highly luminescent nanostructures. Such self-assembly presents the capacity to reduce solvent accessibility, restrict the floppiness of surface ligands, and increase metallophilic interactions through various supramolecular forces. Different types of nanoscale forces such as dipolar interactions, van der Waals interactions, electrostatic interactions, hydrogen bonding, C–H···π interaction, π–π stacking, metallophilic interactions, and amphiphilicity along with the external triggers are responsible for directed self-assembly process of MNCs. The reduction in free energy during the assembly is the main driving force for this process. 

A comprehensive mechanistic understanding of the origins of PL in the self-assembly of NCs is mandatory; in particular, understanding the molecular-level details and structure–property correlation will require a more well-established crystal structure of self-assembly of NCs. The quantum chemistry approach (already well explored by the TD-DFT level of theory in individual MNCs) should bridge the gap to address the photophysical properties of MNCs embedded in self-assembly. In particular, a combined quantum chemical and molecular mechanics method (QM/MM) implementation using periodic boundary conditions might be applied to two-dimensional arrays of metal clusters protected by organic ligands. Such hybrid methods could thus address photophysical processes in the assembly of NCs across length- and time-scales. A combined experimental and theoretical approach including QM/MM tools and computational simulation techniques can provide a holistic description of the nature of the interactions present in self-assembled nanoclusters. QM/MM and molecular dynamic simulations should be developed to simulate self-assembled molecular systems, where an explicit description of changes in the electronic structure is necessary.

Although the emphasis of this review is on the structural correlation of the assemblies with the optical properties, the mechanistic understanding behind the assembly formation is also essential. The successful assembly formation into desired morphology depends on maintaining a balance between the attractive and repulsive forces between the NCs. Concerning the control of self-assembly of MNCs, there is still room for improvement. Indeed, one can take inspiration from recent progress on the fabrication of monodispersed nanoparticles using programmed automated techniques. A possible outcome should be the programmable self-aggregation to activate the AIE process in the assembly of MNCs. The templating motifs with “programmed” surface ligands that drive the assembly formation via different driving forces should be a possibility. Alternatively, the MNC community should learn to create more versatile superstructures with desirable nanoscale optical properties from the metamaterial community. A variety of well-defined MNCs could be obtained through a programmable assembly of NCs. From the scientific perspective, there is a considerable number of reports related to the synthetic chemistry of NCs, whereas the self-assembly of MNCs is not well developed to conclude a relevant mechanism. A wide range of self-assembly processes is reported where NCs are soluble in an organic solvent. In contrast, water-soluble self-assembled NCs are limited due to their lack of stability as crystal structures. However, these water-soluble assembled structures with unique optical properties could be ideal for metal ion sensing, catalysis, bioimaging, and therapeutics [95,122,123]. The assembled structures have been used as biomarkers for cellular imaging and cancer therapy due to their enhanced physicochemical properties, which are advantageous for biological applications [124,125]. MNCs are highly promising in catalysis owing to their tailorable active site structure, which can also tune their catalytic properties. The factors such as size, shape, composition, and isomerization of NCs are also of major concern for new catalysis with high activity and selectivity through site-specific surface tailoring [126,127,128] The selectivity of a catalyst is highly dependent on the morphology of the MNCs [129]. The wide range of emission color tunability of assembled MNCs makes them eligible as a competitive color conversion material in light-emitting diodes (LEDs) [39,41,130]. The advantage of self-assembled structures in device fabrication is due to their decreased surface-to-volume ratio as well as stable tunable emission and their compact arrangement [32]. These veiled built structures could also bring some eccentric optical properties such as AIE, strong circularly polarized luminescence, enhanced mechanochromic properties, etc. Recent progress on fabricating the NCs using templates or linkers is highly appreciated. Aggregated NCs could be confined in a metal-organic framework [84] or zeolite-like framework to maintain their aggregation, restrict the rotation of their ligands, and further improve their QY [131]. Prescribed assembly of inorganic nanoparticles (NPs) guided by the programmable DNA sequences provides a promising method to fabricate various nano-architectures, mainly using DNA origami frame versatility [132]. Finally, simulations can be used to determine the extent of the assembly process and the governing principles for the assembly mechanism.

## Figures and Tables

**Figure 1 nanomaterials-12-00544-f001:**
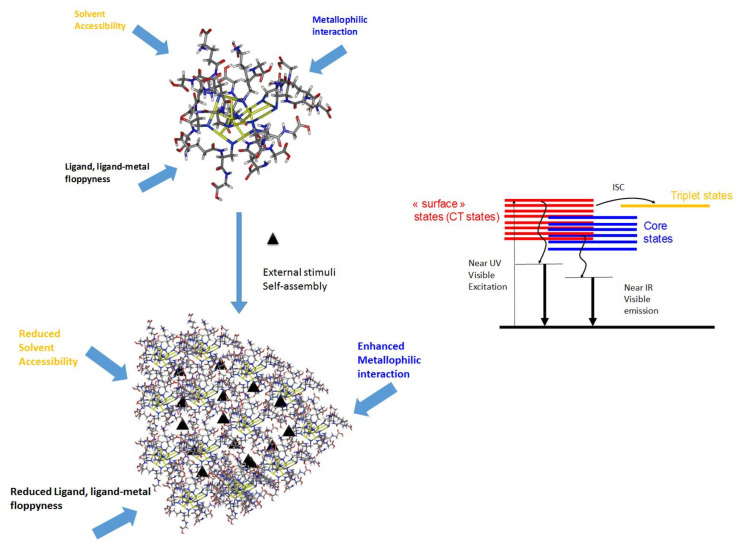
Schematic illustration of the critical parameters affecting the PL properties of MNCs (exemplified with Au_10_SG_10_) and their self-assembly. Proposed mechanism for PL (photoluminescence) in MNCs (metal nanoclusters).

**Figure 2 nanomaterials-12-00544-f002:**
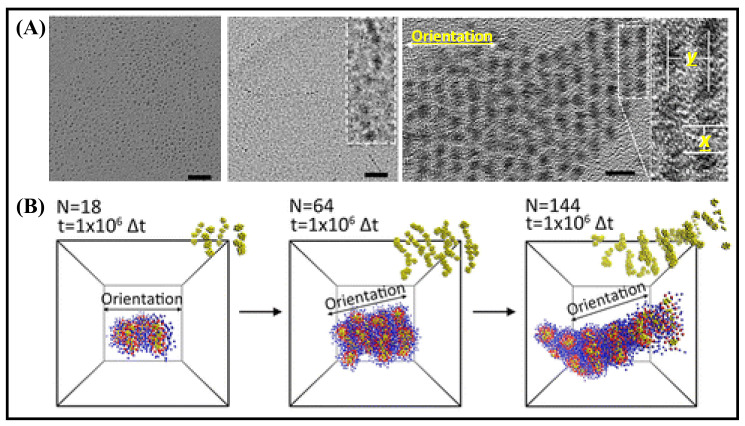
(**A**) TEM images of the morphological evolution of the self-assembled Au15 NCs (a–c), after 1 h at 25 ºC (a), 24 h (b) and at 90 ºC for 5 min (c). The scale bar is 10 nm (a,b) and 5 nm (c). (**B**) Simulation results of the dipole-induced linear arrangement of Au_15_ NCs. It was adapted from Ref. [68], Copyright 2015 American Chemical Society.

**Figure 3 nanomaterials-12-00544-f003:**
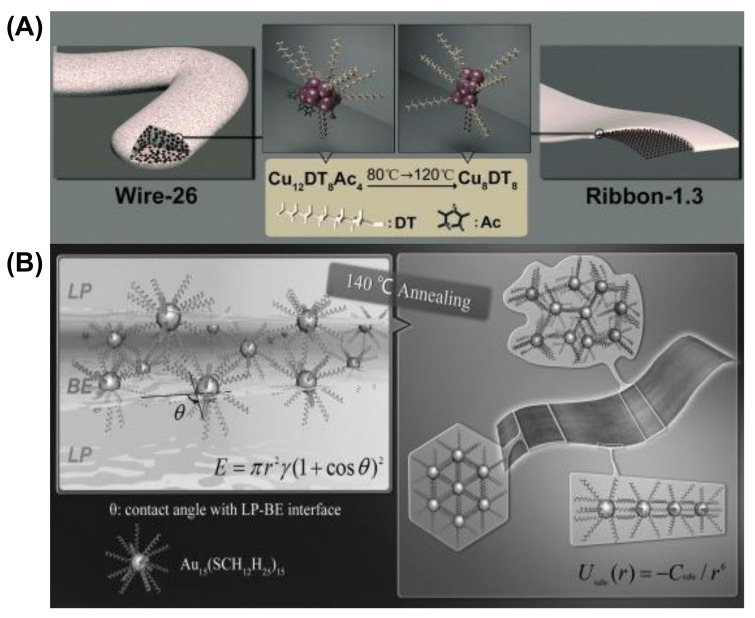
Schematic representation of the morphological evolution of (**A**) the self-assembled Cu NCs from wire-26 into ribbon 1.3. Adapted with permission from Ref. [71]. Copyright 2014 Wiley-VCH Verlag GmbH & Co. KGaA, Weinheim. (**B**) Au_15_ NCs into single-cluster-thick nanosheets. Reprinted from Ref. [72]. with permission. Copyright 2013 Wiley-VCH Verlag GmbH & Co. KGaA, Weinheim.

**Figure 4 nanomaterials-12-00544-f004:**
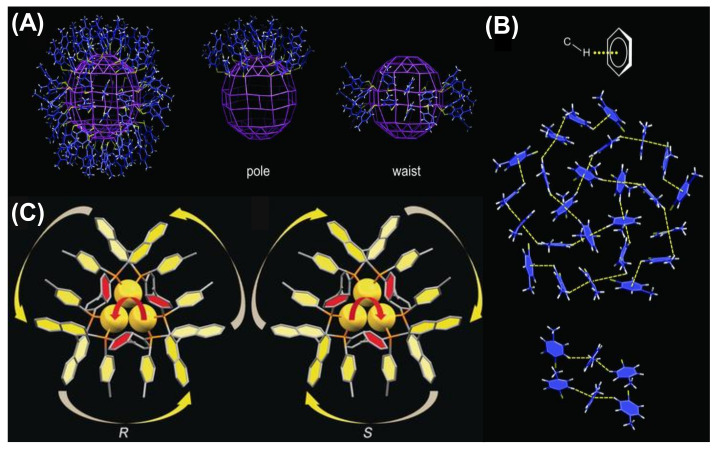
(**A**) The arrangement of the ligands on the surface and (**B**) representation of the C–H···π interactions stabilizing the surface patterns of Au_246_(p-MBT)_80_ NCs. Adapted with permission from Ref. [74]. Copyright 2016, American Association for the Advancement of Science. (**C**) Structural representation of Au_3_[(R)-Tol-BINAP]_3_Cl and Au_3_[(S)-Tol-BINAP]_3_Cl. Adapted from Ref. [75] with permission. Copyright 2017 Wiley-VCH Verlag GmbH & Co. KGaA, Weinheim.

**Figure 5 nanomaterials-12-00544-f005:**
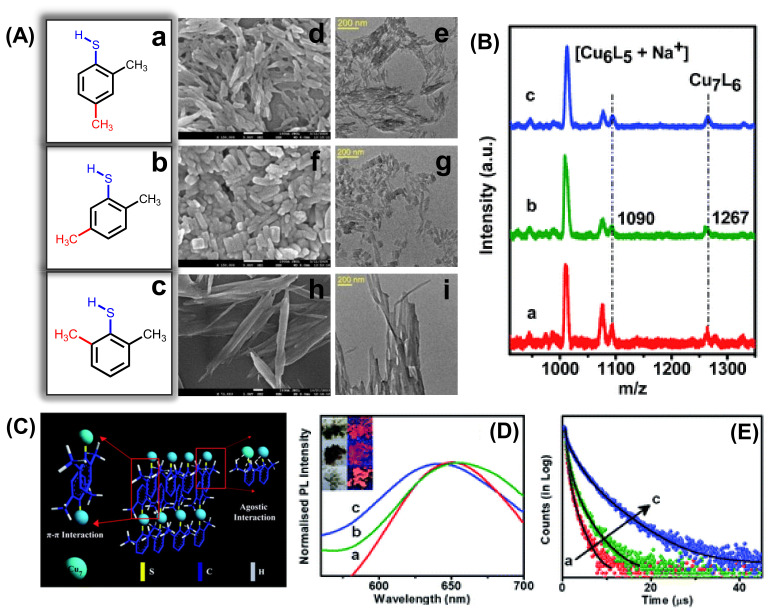
(**A**) The TEM (scale bar is 200 nm) and SEM (scale bar is 100 nm) images of the self-assembled rods, platelets, and ribbon-like structures by varying the isomers of the ligand. (**B**) Structural characterization and (**C**) the evolution process of the assembled NCs. (**D**) Emission spectra and (**E**) PL decay curves of the self-assembled rods (a), platelets (b), and ribbons (c), the inset shows the digital photographs of self-assembled Cu NCs in daylight and under UV (365 nm) excitation. Reprinted with permission from Ref. [76]. Copyright 2021 Royal Society of Chemistry.

**Figure 6 nanomaterials-12-00544-f006:**
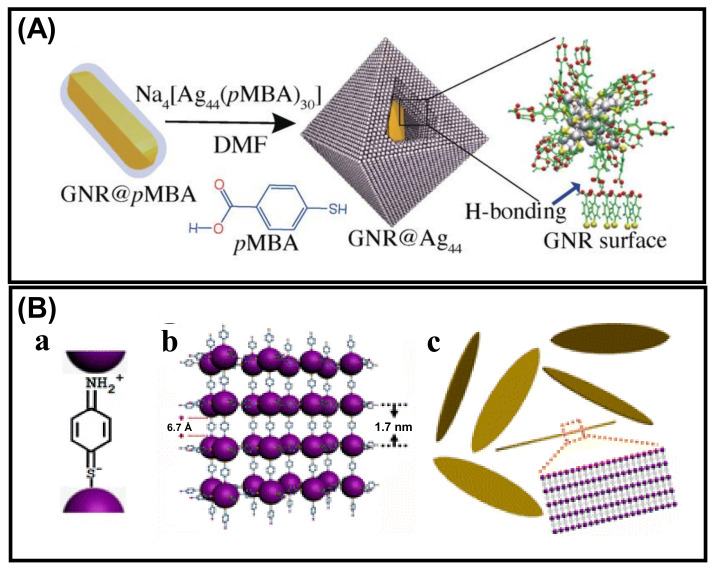
Schematic representation for the (**A**) self-assembly of Ag_44_ NCs on the GNR@p-MBA surface. Adapted with permission from Ref. [78]. Copyright 2018 Wiley-VCH Verlag GmbH &Co. KGaA,Weinheim. (**B**) The quinonoid form of the PATP ligand (a) and formation mechanism of uniform lamellar Ag nano leaves (b,c). Adapted with permission from Ref. [79]. Copyright 2013 American Chemical Society.

**Figure 7 nanomaterials-12-00544-f007:**
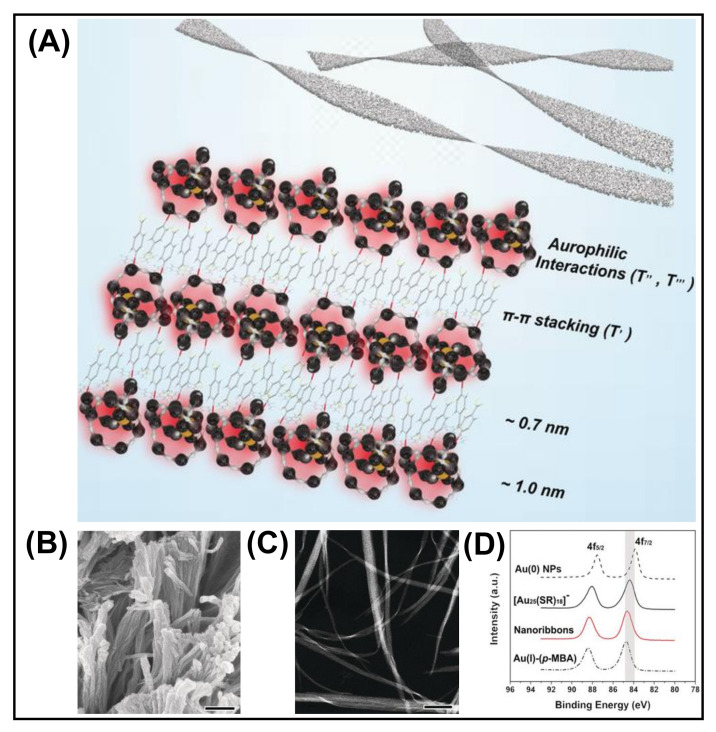
(**A**) Schematic illustration of the aurophilic interaction-directed self-assembly process. (**B**) SEM and (**C**) High-angle dark-field scanning TEM (HAADF-STEM) images of the nanoribbons obtained by the self-assembly of [Au_25_(p-MBA)_18_]^−^ (scale bars are 300 nm for both). (**D**) Au 4f XPS Study of the self-assembled structures. Reproduced from Ref. [82] with permission. Copyright 2019 Wiley-VCH Verlag GmbH & Co. KGaA, Weinheim.

**Figure 8 nanomaterials-12-00544-f008:**
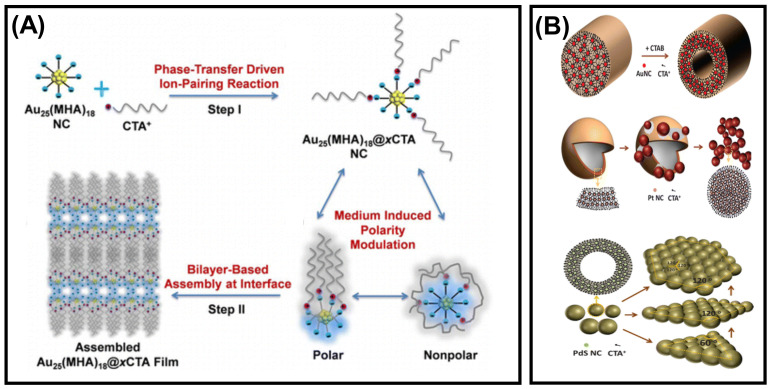
Schematic illustration of the amphiphilicity directed (**A**) self-assembly process of Au_25_(MHA)_18_@xCTA NCs. Adapted with permission from Ref. [86]. Copyright 2015 American Chemical Society. (**B**) Transformation of Au, Pt, and PdS NCs into rod, dandelion, and platelet-like structures. Adapted with permission from Ref. [87]. Copyright 2014 American Chemical Society.

**Figure 9 nanomaterials-12-00544-f009:**
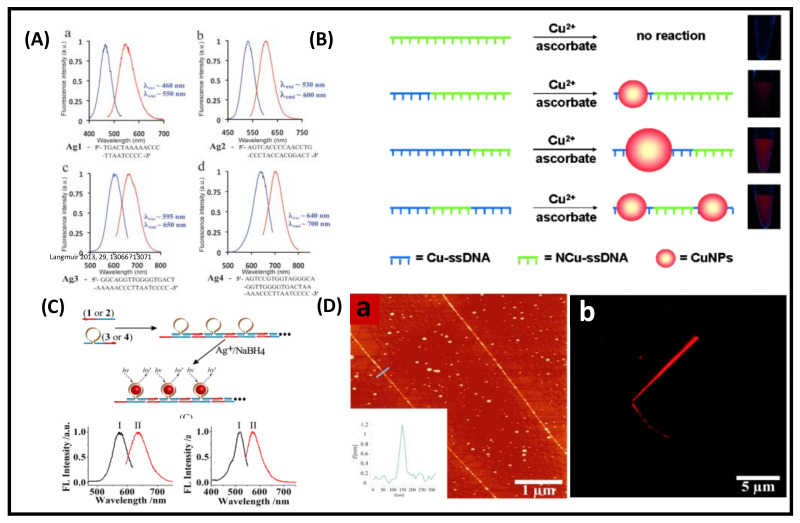
(**A**) Excitation and emission spectra of Ag NCs showing the wavelength-tunable Ag NCs by using different lengths and sequences of DNA. Adapted with permission from ref [90]. Copyright 2010 The Royal Society of Chemistry. (**B**) Cu NPs nanoparticles synthesized by reduction of the ssDNA-Cu^2+^ complex. Cu-ssDNA (blue) represents ssDNA that can serve as a template for Cu NPs, and NCu-ssDNA (green) does not have Cu NP formation capability. The size of Cu NPs depends on the length of Cu-ssDNA. Adapted with permission from ref. [91]. Copyright 2013 John Wiley & Sons, Inc. (**C**) Formation of luminescent Ag NCs by the sequence-specific DNA hairpin structure and excitation (I) and emission (II) spectra of red- and yellow-emitting NCs. (**D**) AFM image of the red emitting Ag NCs (**a**), inset shows the cross-sectional analysis of the self-assembled Ag NCs and confocal microscopy image of wire-like self-assembled Ag NCs with red fluorescence (**b**). Adapted with permission from Ref. [92]. Copyright 2013 American Chemical Society.

**Figure 10 nanomaterials-12-00544-f010:**
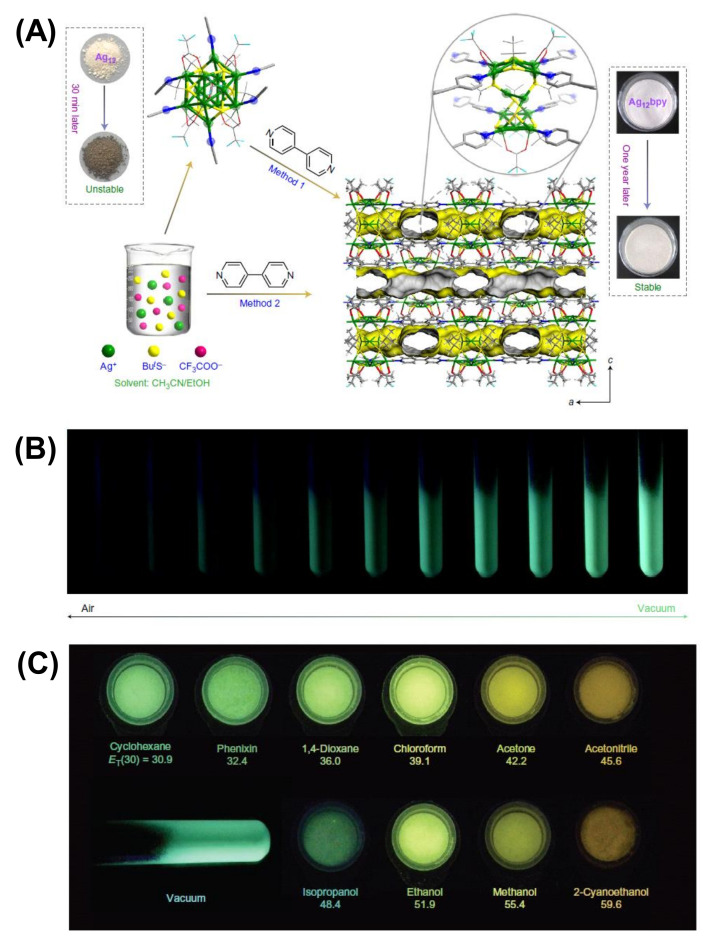
Synthesis and crystal structure of Ag_12_-bpy (**A**), luminescence quenching response to O_2_ (**B**), and vapochromic response of Ag_12_-bpy (**C**). Reproduced with permission from Ref. [95] Copyright 2017 Macmillan Publishers Limited, part of Springer Nature.

**Figure 11 nanomaterials-12-00544-f011:**
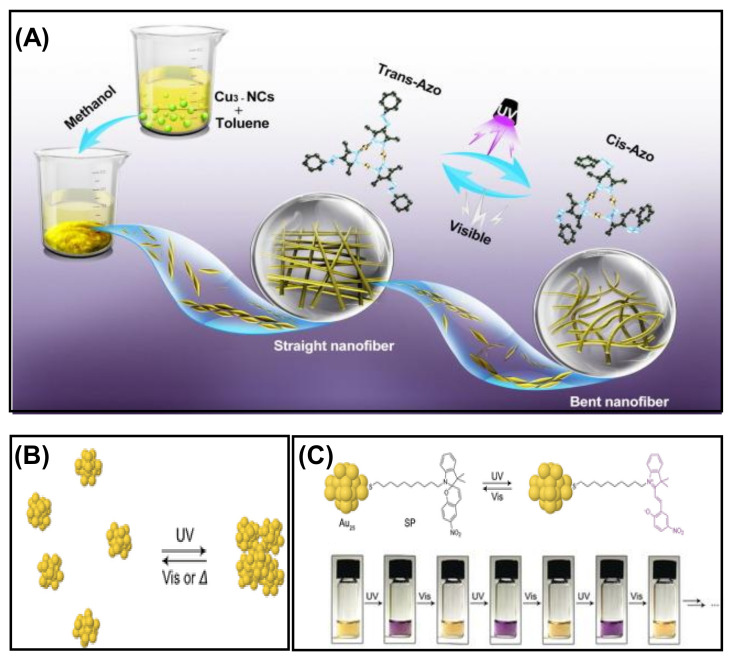
(**A**) Morphological transformation of self-assembled nanofibers triggered by light. Adapted with permission from Ref. [98]. Copyright 2021 Elsevier. (**B**) Schematic representation of the light-controlled self-assembly process. (**C**) The reversible isomerization of spiropyran on Au_25_ NCs in the presence of UV light and digital photographs of Au_25_SP_~3.8_PET_~14.2_ in THF solution upon exposure to UV and visible light alternatively. Reprinted with permission from Ref. [99]. Copyright 2016 Wiley-VCH Verlag GmbH & Co. KGaA, Weinheim.

**Figure 12 nanomaterials-12-00544-f012:**
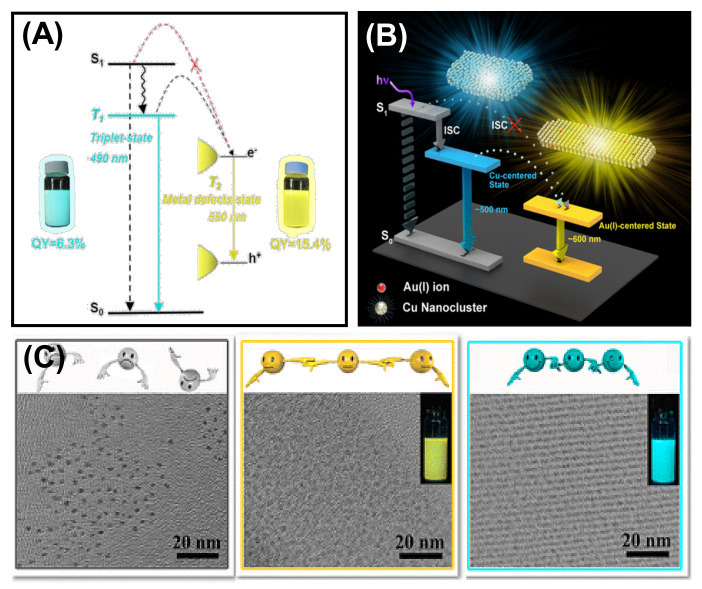
(**A**) Schematic representation of the excited state relaxation dynamics of the Cu NC-self-assembled nanosheets (Adapted with permission from Ref. [112]. Copyright 2017 American Chemical Society) and (**B**) self-assembled nanosheets of Cu NCs with Au doping. Reprinted with permission from Ref. [113]2. Copyright 2017 American Chemical Society. (**C**) TEM images of individual Cu NCs, self-assembled ribbons, and self-assembled sheets of Cu NCs. Inset shows the emission under 365 excitation. Adapted with permission from Ref. [41]. Copyright 2015 American Chemical Society.

**Figure 13 nanomaterials-12-00544-f013:**
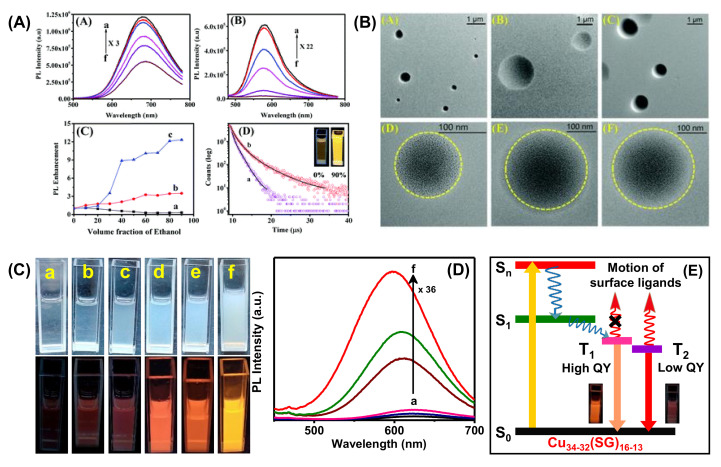
(**A**) Temperature dependent PL spectra of (i) r-AuAg and (ii) y-AuAg NCs, (iii) enhancement of PL for Au and AuAg NCs by changing the volume fraction of ethanol/water (*v*/*v*), (a) Au NCs, (b) r-AuAg, and (c) y-AuAg NCs and (iv) time resolved decay curves of y-AuAg NCs with (a) 0% and (b) 60% ethanol (lex = 415 nm). Inset shows the digital picture of y-AuAg NCs in fv = 0% and fv = 90% under UV-light (365 nm) excitation. (**B**) TEM images of NCs in 60% water–ethanol (*v*/*v*) mixture: (A,D) Au NCs, (B,E) r-AuAg NCs, and (C,F) y-AuAg NCs. Adapted with permission from Ref. [117]. Copyright 2020 The Royal Society of Chemistry. (**C**) Digital photographs of Cu_34-32_(SG)_16-13_ NCs with EtOH volume fraction of (a) 0%, (b) 20%, (c) 40%, (d) 60%, (e) 80%, and (f) 90% under daylight and UV (365 nm) illumination. (D) PL spectra of Cu34-32(SG)16-13 NCs in an EtOH volume fraction of (a) 0%, (b) 20%, (c) 40%, (d) 60%, (e) 80%, and (f) 90%. (**E**) Schematic diagram of excited state dynamics and origin of emission in Cu_34-32_(SG)_16-13_ NCs. Adapted with permission from Ref. [118]. Copyright 2019 American Chemical Society.

## Data Availability

Not applicable.

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
