# Peer review of "Self-Assembled Metal Nanoclusters: Driving Forces and Structural Correlation with Optical Properties"

_nanomaterials, 2022, doi:10.3390/nano12030544_

Round 1

Reviewer 1 Report

This manuscript gives a comprehensive review on metal nanoclusters especially focusing on the structure-optical properties relationships and their self-assemblies, which have been described only in a limited scope in literatures for this class of nanomaterials. Firstly, nanoscale forces (such as dipolar interactions, vdW interactions, electrostatic interactions, and metallophillic interactions) are provided as a fundamental basis to understand the self-assembly of functionalized metal nanoclusters with satisfactory literature examples. Then methodologies for self-assembly (such as template-directed assembly, DNA template-directed self assembly, and linker-directed assembly, and light-triggered self-assembly) are shown with suitable precedents. Finally, optical properties are described for the self-assembled nanoclusters in the context of aggregation-induced emission. This review will provide a nice and helpful clue to chemistry and materials science of metal nanoclusters. 

Author Response

Authors' reply: Thank you very much for appreciating our effort.

Reviewer 2 Report

“Structure-Optical Properties Relationship and 3 Driving Forces Involved in the Self-Assembly of 4 Metal Nanoclusters” by  Sarita Kolay et al.

 I went through the paper very carefully and thoroughly. Authors studied the self-assembly of metal nanoclusters (MNCs) which are an emerging field of research owing to their significant optical properties and potential applications in many areas. Fabricating the desired self-assembly structure for specific implementation has always been challenging in nanotechnology.

1-The paper contains interesting sciences in PCs. The impact of the paper on optical properties of metal nanoclusters is going be good. Also, the quality of the research work presented in the paper is also good. .

2-In general, ideas are well explained and understandable but, some tenses, linkers and grammar structures must be checked.

3-The authors should give the thickness and number of layers of all layers that they calculated. Are these parameters obtained from an optimization process?

  1. What is the novelty of this manuscript compared to published results?
  2. The authors should argue about the relevance of the temperature dependence of the coating.
  3. The Introduction does not provide sufficient background. The introduction does not explain the major contributions and novelty of this work. The significance of the proposed solution has not been summed up.
  4. There are someequations without references?

8- The constructive discussions are missing. As mentioned earlier, authors must make a comparative analysis with other similar solutions and back up their claims on how the proposed solution can be considered as high performing compared to others

9- How their results will be affected if they include energy loss in layers.

10- The novelty of this work should be stated explicitly in the text of the manuscript so that readers can get it easily.

11- Authors should compare their results with the published data and different results.

12- Authors mentioned in the conclusion” Finally, simulations can be used to determine the extent of the assembly process and the governing principles for the assembly mechanism”. The significance of the simulated results is high, when these modeling (simulation) satisfactory agrees with the experimental data. Nonetheless, the comparison of the simulated and experimental results is desirable.

13- Authors should explain one or two application to their work.

14- All figures, symbols, equations should be improved.

15- It seems the title need revision by authors to become more informative

16- Authors mentioned “Even if the detailed photoluminescence (PL) mechanism in gold NCs is incomplete………………………………” this sentence need more explain and evidence

17- Authors mentioned” Solvent evaporation is enhanced after annealing at 140 ……………..” this kind of sentences need confirmation by reference?

18- Table one need more revision, especially symbols

19- Authors mentioned” A combined quantum chemical and molecular mechanics method (QM/MM) implementation using periodic boundary conditions might be applied to two-dimensional arrays of metal clusters protected by organic ligands. Such hybrid methods could thus address photophysical processes in assembly of NCs across length- and timescale” How did this combination?

20-Finally, I recommend that the paper should be revised taking care of the above comments.

I wish to resend this paper after corrections and revise my comments

Author Response

Comments and Suggestions for Authors

"Structure-Optical Properties Relationship and 3 Driving Forces Involved in the Self-Assembly of 4 Metal Nanoclusters" by Sarita Kolay et al.

I went through the paper very carefully and thoroughly. Authors studied the self-assembly of metal nanoclusters (MNCs) which are an emerging field of research owing to their significant optical properties and potential applications in many areas. Fabricating the desired self-assembly structure for specific implementation has always been challenging in nanotechnology.

1-The paper contains interesting sciences in PCs. The impact of the paper on optical properties of metal nanoclusters is going be good. Also, the quality of the research work presented in the paper is also good.

Authors' reply: Thank you very much for appreciating our effort.

2-In general, ideas are well explained and understandable but, some tenses, linkers and grammar structures must be checked.

Authors' reply: Thank you for your suggestions. We have checked the manuscript and corrected tenses, linkers, and grammar structures. We appreciate the important point raised by the reviewer and tried our best to address the following questions.

3-The authors should give the thickness and number of layers of all layers that they calculated. Are these parameters obtained from an optimization process?

Authors' reply: Thank you for your comments. We have given the thickness and number of layers from the literature survey and provided proper references (Angew. Chem. Int. Ed. 2014, 53, 12196 –12200). The thickness value given in the manuscript is calculated from an atomic force microscopy study. (Angew. Chem. Int. Ed. 2014, 53, 12196 –12200)

4-What is the novelty of this manuscript compared to published results?

Authors' reply: Thank you for your comments. Although few review papers are published on self-assembly of metal nanoclusters (MNCs), those are focused on the design principles and governing chemistry of the self-assembly. Also, their potential applications in various field have been discussed in the previously published review (ACS Mater. Lett. 2019, 1, 237-248, Small 2021, 17, 2005718). The structural correlation of the MNCs' optical properties is significant to design the self-assembled structures with customizable properties. However, the detailed study about the mechanism is not highlighted yet. Thus, we focused mainly on correlating the assembled structures with their optical properties in this review. It will help us prepare the MNCs with proper tunable emission property for specific applications.*

The following paragraph has been added at the end of Introduction:

“There are many recent reviews on the synthetic strategies to produce self-assembly of NCs and their applications.[50-59] Whereas, the structural correlation of the MNCs with their optical properties remain unveiled which is also important to design the self-assembled structures with customizable properties. Here, in this review we particularly will focus on the self-assembled structure-optical properties relationship and the driving forces behind assembly formation.”

5-The authors should argue about the relevance of the temperature dependence of the coating.

Authors' reply: Thank you very much for this suggestion. With changing the temperature, self-assembled geometry also changes due to the reorganization of the MNCs, which we have explained in detail in the modified manuscript (page 6 in the revised manuscript). (Angew. Chem. Int. Ed. 2014, 53, 12196 –12200) Generally, with the elevation of temperature, surface ligands suffer more vibrational motion, which significantly enhances the non-radiative channels and affects the photophysical properties of the assembled structure.

6-The introduction does not provide sufficient background. The introduction does not explain the major contributions and novelty of this work. The significance of the proposed solution has not been summed up.

Authors' reply: Thank you for your comments. In this review, we have briefly discussed the driving forces behind the assembly formation of the NCs and the structural correlation with their optical properties. All the significant contributions in these fields are discussed separately under proper subtitles. Thus, the introduction provides a general overview of the self-assembly phenomenon of MNCs, optical properties, driving forces behind the assembly formation, etc. However, we have added some information with proper references as per your suggestion. (Page 2 & 25 in the revised manuscript) Coming to your next concern about the uniqueness of this work, although there are few review papers published on self-assembly of MNCs, those are focused on the design principles and governing chemistry of the self-assembly (ACS Mater. Lett. 2019, 1, 237-248, Small 2021, 17, 2005718). Also, their potential applications in various fields have been discussed in the previously published review (ACS Mater. Lett. 2019, 1, 237-248, Small 2021, 17, 2005718). The structural correlation of the  MNCs' optical properties is significant to design the self-assembled structures with customizable properties. However, no detailed discussion about the mechanism is given in the previous literature. Thus, we focused mainly on correlating the assembled structures with their optical properties in this review. It will help us to prepare the MNCs with proper tunable emission property for specific applications.

The following paragraph has been added at the end of Introduction:

“There are many recent reviews on the synthetic strategies to produce self-assembly of NCs and their applications.[50-59] Whereas, the structural correlation of the MNCs with their optical properties remain unveiled which is also important to design the self-assembled structures with customizable properties. Here, in this review we particularly will focus on the self-assembled structure-optical properties relationship and the driving forces behind assembly formation.”

7-There are some equations without references?

Authors' reply: Thank you for pointing out our mistakes. We have included the references for all equations. (Page 5 & 6 in revised manuscript).

8- The constructive discussions are missing. As mentioned earlier, authors must make a comparative analysis with other similar solutions and back up their claims on how the proposed solution can be considered as high performing compared to others

Authors' reply: Thank you for your comments. In this review article, we have discussed the self-assembly of MNCs, the driving forces behind them, and their optical properties. Similar review papers in these fields mainly focus on nanoscale forces and their application. Whereas, we have tried to briefly focus on the structural correlation of the assembled MNCs with their optical properties, which is unique compared to others. (see section 2. On the Structure-Optical Properties Relationship of Metal Nanocluster Assembly).

9- How their results will be affected if they include energy loss in layers.

Authors' reply: Thank you for your comments. In this review, we have included a few pieces of literature that correlate the excited state energy loss process with the assembled structures (J. Phys. Chem. C 2019, 123, 2506-2515, ACS Appl. Mater. Interfaces 2017, 9, 24899-24907). In general, the greater extent of inter-NC binding gives rise to more compact assembled structures, reducing the non-radiative energy loss during excited state decay processes. So, the compactness of assembly increases PL (J. Am. Chem. Soc. 2015, 137, 8244-8250, Nanoscale Adv. 2021, 3, 5570-5575). This is a general phenomenon in self-assembled structures. We are highlighting the driving forces for the formation of self-assemblies.

10- The novelty of this work should be stated explicitly in the text of the manuscript so that readers can get it easily.

Authors' reply: Thank you very much for your comments. Although few review papers are published on self-assembly of MNCs, those are focused on the design principles, governing chemistry of the self-assembly, and their applications. (ACS Mater. Lett. 2019, 1, 237-248, Small 2021, 17, 2005718) The structural correlation of the MNCs' optical properties is very important to design self-assembled structures with customizable properties. This review focuses on correlating the assembled structures with their optical properties. It will help us to prepare the MNCs with proper tunable emission property for specific applications. (Page 2 in the revised manuscript)

11- Authors should compare their results with the published data and different results.

Authors' reply: Thank you for your comments. We have discussed the self-assembly of MNCs and the underlying chemistry behind this formation. We have tried to review all the significant contributions of this area and briefly discussed them.

12- Authors mentioned in the conclusion" Finally, simulations can be used to determine the extent of the assembly process and the governing principles for the assembly mechanism". The significance of the simulated results is high, when these modeling (simulation) satisfactory agrees with the experimental data. Nonetheless, the comparison of the simulated and experimental results is desirable.

Authors' reply:  Thank you for this cogent issue. The following paragraph has been add in conclusion : “A combined experimental and theoretical approach including QM/MM tools and computational simulation techniques (molecular dynamics) can provide a holistic description of the nature of the interactions present in self-assembled nanoclusters.”

13- Authors should explain one or two application to their work.

Authors' reply: Thank you for your comments. This review article focused on the structural correlation of MNCs with their optical properties. (ACS Mater. Lett. 2019, 1, 237-248, Small 2021, 17, 2005718) So, we have discussed the driving forces behind the assembly formation and the impact of assembly structure on photophysical properties. Self-assembled MNCs have extensive applications in white light-emitting diodes, catalysis etc., which is mentioned in the introduction part. However, explaining their applications would require a detailed literature survey, which we may cover in the next review article. Although as per your suggestion, we have briefly discussed the scope of applications with the assembled structure (Page 25 in the revised manuscript).

14- All figures, symbols, equations should be improved.

Authors' reply: Thank you for your comments. We have tried to modify the figures' quality and symbols used in equations and hope we have justified the reviewer's concern. (Figure 6 and Figure 13 have been improved)

15- It seems the title need revision by authors to become more informative

Authors' reply: Thank you very much for your suggestion. Here is a more informative new title : “Self-assembled Metal Nanoclusters: Driving Forces and Structural Correlation with the Optical Properties”.

16- Authors mentioned "Even if the detailed photoluminescence (PL) mechanism in gold NCs is incomplete………………………………" this sentence need more explain and evidence

Authors' reply:  Definitely this sentence required more explain and evidence. Here is the revised paragraph (beginning of Section. 2):

“The origin of the photoluminescence (PL), in particular in the near-infrared, from thiolate-protected gold nanoclusters remains elusive. Indeed, it is still a major challenge for researchers to map out a definitive relationship between the atomic structure and the PL property and understand how the metal core (through excitations through Au(0) kernel) and Au(I)–S surface (through charge transfer excitations) contribute to the PL of Au NCs.”

17- Authors mentioned" Solvent evaporation is enhanced after annealing at 140 …………….." this kind of sentences need confirmation by reference?

Authors' reply: Thank you for your suggestion. We have provided the reference for the sentence (Page 7 in the revised manuscript, ref no. 72).

18- Table one need more revision, especially symbols

Authors' reply: Thank you for your comments. We have tried to improve the abbreviation and hope we have justified your concern.

19- Authors mentioned" A combined quantum chemical and molecular mechanics method (QM/MM) implementation using periodic boundary conditions might be applied to two-dimensional arrays of metal clusters protected by organic ligands. Such hybrid methods could thus address photophysical processes in assembly of NCs across length-and timescale" How did this combination?

Authors' reply: Quantum mechanics/molecular mechanics (QM/MM) molecular dynamics (MD) simulations should be developed to simulate self-assembled molecular systems, where an explicit description of changes in the electronic structure is necessary. This sentence was added in conclusion.

20-Finally, I recommend that the paper should be revised taking care of the above comments. I wish to resend this paper after corrections and revise my comments

Round 2

Reviewer 2 Report

Authors considered all my comments and it seems the updated version are ok